# Use of Fluidized Bed Combustion Fly Ash as a Partial Substitute for Cement in Underwater Concrete Mixes

**DOI:** 10.3390/ma15144809

**Published:** 2022-07-10

**Authors:** Elżbieta Horszczaruk, Cyprian Seul

**Affiliations:** Department of Civil and Environmental Engineering, West Pomeranian University of Technology in Szczecin, al. Piastów 50a, 70-311 Szczecin, Poland; cyprian.seul@zut.edu.pl

**Keywords:** cement, fluidized bed combustion fly ash, underwater concrete

## Abstract

Despite limitations to coal combustion energy production, many countries face the still-unresolved problem of utilising the wastes from fluidised bed coal combustion. One direction of rational utilisation can be using these wastes in the building materials industry. The study aimed to analyse the possibility of using fluidised bed combustion fly ashes as a partial substitute for cement in the underwater concrete (UWC). Two groups of concrete mixes were tested, containing 20 to 50% of fluidised bed combustion fly ashes. Investigations of the rheological properties of the concrete mixes and the mechanical performance of the hardened concrete confirmed the possibility of replacing cement in UWC with fluidised bed combustion fly ash up to 30% of the cement mass. The higher content of the fly ashes significantly worsens the UWC strength as well as the consistency and wash-out loss of the concrete mixes, excluding its use in underwater concreting.

## 1. Introduction

Countries with mainly coal-based energetics face problems utilizing the large volume of fly ashes, the by-products of combustion. In recent years, Poland has become Europe’s largest producer of coal [1]. In 2020, Poland overtook Germany in using coal, which delivers 74% of the country’s electric energy. As a result, more than 4.2 Mt of fly ashes are produced in Poland by combusting fossil fuels or their co-combustion with biomass [2]. Only a small portion of these fly ashes (<10%) are recycled, particularly in construction [3,4,5,6]. More effective bed fluidized combustion systems have been introduced to improve the energetic efficiency. In many countries, including Poland, a systematic increase in the production of by-products from fluidized bed coal combustion has been observed [7].

Fluidized bed combustion (FBC) fly ashes have a different chemical composition than the fly ashes from conventional coal dust combustion. The difference is a result of the lower temperature of FBC, which is 800–900 °C, compared to conventional beds, in which the coal is combusted at 1400–1700 °C [7,8]. During FBC, the mineral components of the fuel are transformed. At a temperature above 500÷550 °C, dehydroxylated clay minerals are decomposed, creating amorphic aluminosilicates with relatively large specific surface area [9]. The desulfurization products are usually totally oxidized and occur most often in the form of anhydrite [10]. The presence of large amounts of dehydroxylated clay minerals is significant for the pozzolanic activity of these wastes (they quickly react with calcium hydroxide). These materials reach high reactivity resulting from thermal activation because, during roasting at the proper temperature, the hydroxylic groups are removed from the structure of the clay minerals [11]. These structural changes are significant enough to lead to amorphic phase formation, which significantly improves the pozzolanic activity of the fluidized bed combustion fly ashes [12,13]. FBC fly ashes contain no glass phase [14], in contrast to siliceous fly ashes.

The physico-chemical properties of fluidized bed combustion fly ashes (FBCFAs) depend mainly on the combusted fuel and combustion technology [15,16,17].

Large amounts of sulfates, mainly the anhydrite, and free calcium oxide in the fluidized bed combustion fly ashes limit their use in the cement composites [18,19,20]. The fluidized bed combustion fly ashes do not meet the European Standards’ requirements for additions to cement [21] and concrete components [22]. In Poland, according to the Standard [21], the fluidized bed combustion fly ash may be used up to the content of 5% as a secondary mineral addition for cement production. As a concrete component, it is used only on the base of the Technical Approvals. Some FBCFA do not meet the requirements of the Standard’s classification for the class C or F fly ash [23]. In some countries, using the fluidized bed combustion fly ashes is standardized and thus allowed. An example is Czech Standard [24], allowing for broad use of FBC fly ashes, also as an addition to concrete.

The introduction of still new systems of the fluidized bed combustion not only leads to the limitation of CO_2_ and heavy metals emission but also causes some stabilization of the chemical composition of the combustion products, including fly ashes. Investigations of new types of FBC fly ashes are intensively conducted in the countries facing a significant problem with re-using the rising amount of FBCFAs. The ashes are utilized to neutralize the acidic soil components, to fertilize soil, to improve water transport in the soil, to limit fertilizers and fine-soil-fractions wash-out [25,26]. An important direction of FBCFA utilization is broadly understood to be environmental protection. Due to the aforementioned high content of the clay minerals, FBCFAs can be used for producing the insulation layer in communal or industrial waste landfills. The high pH value of fluidized bed combustion fly ashes makes it possible to use them to neutralize the acidic leakages from landfills [20]. These fly ashes are also largely used in road construction and road infrastructure engineering. They can be utilized as a material for the construction of embankments, a filler for bituminous mixes, a material for anti-filter screens, a substrate stabilizer, and a material for making the structural layers of roads [4,7]. Cement-based building materials, as well as the cement itself, are still the primary direction of research on the potential use of FBCFA [5,12,27,28]. Additionally, successful investigations on the production of the cementless binders containing FBC fly ashes are being carried out, focusing mainly on the alkali-activated binders, the so-called geopolymers [29,30,31]. However, there is little research into the use of fluidized bed combustion fly ashes in the underwater concrete (UWC) mixes [32,33,34]. Therefore, a trial of more comprehensive analysis of the results obtained so far has been undertaken.

The paper aims to demonstrate the possibility of using FBC fly ash in underwater concrete mixes. The concrete mixes with different water to binder ratios w/b (0.40 and 0.46) were analyzed in the presented investigations. The percentage of FBCFA ranged from 20 to 50% of the cement mass. The investigation covered 10 underwater concrete compositions. The compressive strength of the concrete after 28 and 56 days of underwater curing was determined. The penetration depth of the water under pressure was tested after 28 days of underwater curing. The consistency and wash-out loss of the UWC mixes were determined, and the results were analyzed from the point of view of the underwater concreting technology (e.g., tremie method or pumping method). 

## 2. Materials and Methods

### 2.1. Materials and Specimens

The following materials were used to prepare the tested UWC: Portland cement CEM I 42.5N with the specific density of 3.1 g/cm^3^, gravel of 2–8 mm and 8–16 mm fractions, natural sand 0–2 mm and tap water. The grain size distribution curves of the sand and gravel are presented in Table 1. A polycarboxylate superplasticizer (SP) was added in variable amounts to obtain the slump of at least 400 mm, recommended in the German Standard DIN 1045-2 [35] for UWC mixes. The anti-washout admixture (AWA) containing synthetic cellulose ethers, counted into the group of the stabilizing admixtures and dedicated specifically to underwater concrete, was also used to limit the wash-out loss. The fly ash from the combustion of the black coal in circulating fluidized beds in the Żerań heat and power plant (Warsaw, Poland) was used as a partial substitute for the binder in the tested UWC. The density of the fly ash was 2.54 g/cm^3^. The chemical composition of the used FBCFA and cement are presented in Table 2. The particle size distribution of FBCFA is presented in Figure 1.

Two series of concrete mixes were designed with the water to binder ratio w/b equal to 0.4 and 0.48 and with variable content of cement and superplasticizer. The amount of AWA was adjusted proportionally to the binder content, 1% of the binder mass. The concrete mixes contained different volumes of FBCFA, which was added as a partial substitute for the cement. The concrete mixes’ compositions are presented in Table 3. The specimen designation CRX-Y means: R—reference concrete without FBCFA; X—w/b ratio; Y—the percentage of FBCFA in the binder. 

The UWC mixes were prepared in the laboratory mixer with 30 L. The concrete specimens were prepared in cubic steel molds 100 × 100 × 100 mm. The molds were first placed at the bottom of the container, which was then filled with water until the water level was 20 cm above the upper edge of the mold. Then, the concrete mix was cast directly from above the water table without smoothing the formed surface of the concrete (Figure 2). The specimens were demolded after two days and then stored in the water containers at 18–20 °C until compressive strength testing. Immediately before the test, the specimen’s top surface was levelled using a concrete saw. The compressive strength was determined after 28 and 56 days of specimen curing.

### 2.2. The Procedure of Determination of UWC Mixes Wash-Out Loss

The method for determining wash-out loss for the concrete mix has been developed in Belgium. The method has been standardized in the USA and described in the instruction CRD-C61-89A [36]. The procedure consists of immersing the concrete mix sample, placed in the perforated container, in the cylindrical plastic pipe filled with water. Then, the loss of mass of the concrete mix is determined. The concrete mix container is cylindrical with a diameter of 130 ± 2 mm and a height of 120 ± 2 mm. It consists of a perforated steel plate with a thickness of 1.4 mm (Figure 3). The transparent pipe in which the test is performed has a height of 2000 ± 2 mm and an internal diameter of 190 ± 2 mm (Figure 4). 

The determination of the concrete mix wash-out loss was carried out as follows:The pipe was filled with water up to the level of 1700 ± 5 mm.The container with the cover was weighed, and the concrete mix sample with a mass slightly exceeding 2000 g was placed in the container.The concrete mix in the container was compacted using the metal rod, and the container walls were cleaned from the leaking concrete mix. The mass of the concrete mix (M_i_) was again measured; it should be 2000 ± 20 g.The container with the concrete mix was attached to the cable and placed above the water surface. Then, the container was freely lowered to the bottom of the pipe.After 15 s, after the container had fallen to the bottom, it was pulled out in a time not longer than 5 ± 1 s. The excess water was removed from the container, leaving it to drain for 2 min, and then the container was weighed (M_f_).The described procedure was conducted three times for each concrete mix sample.

The loss of mass of the concrete mix sample was the difference between M_i_ and M_f_. For each sample, a percentage of mass loss D (the so-called wash-out loss) was calculated after each test. The mass loss D after the third immersion was accepted as the final result. The wash-out loss of UWC mixes were determined immediately after mixing the components and one hour after the mixing in the laboratory mixer.

### 2.3. Methods—Consistency Test

The consistency of the UWC mixes was determined using the flow table according to the European Standard EN 12350-5 [37]. The test was performed for each concrete mix immediately after mixing the components and one hour after the mixing.

### 2.4. Methods—Compressive Strength and Watertightness of Concretes

The concrete specimens’ compressive strength was determined according to the European Standard EN 12390-3 [38] after 28 and 56 days of curing the specimens in the conditions described in Section 2.1. Six specimens were tested for each series of concrete. The tests were performed using the strength machine with a maximum pressure of 5000 kN. The rate of increasing the compressive stress was 0.2 to 1.0 MPa/s.

The watertightness of the concrete was determined by testing the depth of water penetration under pressure. The test was determined according to EN 12390-8 [39]. The tests were conducted on the cube specimens with dimensions 100 × 100 × 100 mm. After demolding the specimens, the surfaces the water shall penetrate have been ground. The tests started after 28 days of specimens curing. The water pressure of 0.5 MPa was applied and kept for 72 h. Then, the specimens were split in two in the strength machine in a direction perpendicular to the penetrated surface. After splitting, the range of water penetration was measured with an accuracy of 1 mm. The final result was the maximum depth of water penetration measured for the specimens of the given series.

## 3. Test Results and Analysis

### 3.1. Properties of UWC Mixes

The results of UWC mixes consistency testing, according to EN 12350-5 [37], are presented in Figure 5. According to the American recommendations [40], the flow diameter of UWC mixes after 60 min should exceed 255 mm. All tested concrete mixes have demonstrated the required consistency. 

The effect of the binder content on the tested concrete mixes’ consistency was visible. The UWC mixes with the binder content of 400 kg/m^3^ and w/b = 0.48 had the consistency of SCC mix or close to it. All concrete mixes from the C0.48 series obtained flow diameters above 400 mm after 0 and 60 min. Such a flow is recommended for UWC mixes in the German Standard DIN 1045-2 [35]. However, no recommendations regarding the time of consistency testing are given in this standard. For the concrete mixes from the C0.4 series, with a high binder content of 530 kg/m^3^, the flow diameter above 400 mm was noted after mixing the components of the concrete mix CR0.4-0 without fly ash addition. Despite using a large amount of superplasticizer SP, the strong influence of FBCFA content on consistency of the tested UWC mixes was observed. This effect was particularly significant when the FBCFA content exceeded 30% of the cement mass. The used volume of SP was very high in these concrete mixes, exceeding even 3% of the binder mass, and significantly affected the UWC mixes’ production cost. Only slight changes in the consistency with time were observed in the case of the UWC mixes with w/b = 0.48. The reason for this could be the properly adjusted SP content in this series of concrete mixes. In the case of the mixes with w/b = 0.4, when the content of FBCFA increased, the rise of water demand led to a significant worsening of workability and consistency despite the high amount of SP admixture. The results of consistency testing are presented in Table 4.

The results of UWC mixes’ wash-out loss testing according to CRD-C61-89A [36] are presented in Figure 6. The wash-out loss D was determined after 0 and 60 min of mixing components. The content of AWA was constant and equal to 1% of the binder mass in all concrete mixes. The determined wash-out losses for the individual UWC mixes are presented in Table 4.

The wash-out loss D was increasing with the growth of FBCFA content in the tested UWC mixes. An effect of AWA content on D value was visible after 60 min. from the mixing of the components. This effect was demonstrated by decreasing the D value compared to the values obtained immediately after mixing components. According to the American recommendation [40], the wash-out loss after 60 min. should not exceed 12% for the ordinary concrete mixes and 8% for the high-strength concrete mixes. Only two tested concrete mixes with w/b = 0.40 demonstrated the wash-out losses below 12% after 60 min. (the concrete mixes CR0.4-0 and CR0.4-20). The high contents of FBCFA and high total volume of the binder in the concrete mixes with w/b = 0.4 caused the water amount to be too low for hydration, and the binder was washed out during concreting under water. The above-described mechanism explains why high wash-out loss values were observed even after 60 min. from the mixing. All UWC mixes with w/b = 0.48 obtained wash-out losses below 12%. D values for the concrete mixes of the C0.48 series rose with the FBCFA content, but after 60 min., they still did not exceed 8%. Together with maintaining the consistency required by the document [40] (see Figure 5), it can ensure the proper placing of concrete mix in the formwork under water at the high reinforcement volume. 

### 3.2. Compressive Strength

The UWC compressive strength testing results after 28 and 56 days of curing are presented in Table 5 and Figure 7. A slight increase in compressive strength was observed for the concrete with w/b = 0.40 and FBCFA content 20% and 30% compared to concrete CR0.4-0 (without FBCFA); it was 3.2% and 1.7%, respectively, after 28 days, and 9.5% and 2.6 %, respectively, after 56 days. A decrease in compressive strength has been noted for concrete C0.4-40 and C0.4-50; it was 6.7% and 8.6%, respectively, after 28 days, and 0.4% and 5.4%, respectively, after 56 days. Slightly diminishing strength for concrete with w/b = 0.48 has been observed at 30% content of FBCFA, while maximum falls were noted at 50% content of FBCFA and were equal to 16.8% and 18.2% after 28 and 56 days of curing, respectively. The found rise of compressive strength for UWC containing up to 20% of FBCFA was a result of sealing the tested concrete’s microstructure. As presented in [13,41,42], an addition of fly ash caused growth of the volume of crystalline ettringite inside the cement matrix, leading to an increase in the compressive strength of concrete. When FBCFA content exceeded 30%, UWC compressive strength decreased. The diminishing was milder with concrete ageing. The rate of strength changed depending on the binder content in the concrete mix. 

The high binder content in the UWC mix generated more considerable wash-out loss and worsening of the mix’s workability, which can finally increase the volume of the micropores and macropores in concrete. A lack of sufficient water to wet the high volume of the fly ash can also result in a fall in the compressive strength. According to [43], increasing the content of the fluidized bed combustion fly ashes causes growth of the pores’ distribution index and a decrease in the specific surface area of the pores.

### 3.3. Depth of Water Penetration under Pressure Watertightness of the Concretes

The results of testing water penetration depth are presented in Table 6. The UWC specimens were subject to the hydrostatic pressure of 0.5 MPa for 72 h. The water penetration depth was inversely proportional to FBCFA content up to 30% of the cement mass. Above this value, only a slight increase in water penetration depth was observed; for both series of tested concrete mixes, it did not exceed the values obtained for the reference concrete compositions CR0.4-0 and CR0.48-0 (without fly ash addition). The research presented in [44] has confirmed that adding FBCFA in the amount of up to 30% of the cement mass causes a limitation in chlorides penetration into concrete, improving the resistance of the composite to reinforcement corrosion.

The initial increase in the tightness of concrete was a result of concrete’s structure compaction in the range of the capillary pores by FBCFA microparticles. As presented in [13], an addition of 20–30% of FBCFA caused a rise in the C-S-H phase and crystalline ettringite volume, which was evidenced by the growth of the amount of water bound in the hydration products and fall in the portlandite volume. As reported in [43], the specific surface area of the pores diminished with the increase in FBCFA content (above 20%). The above means that pores with larger diameters were created. The pore size distribution analysis presented in [43] confirmed that high FBCFA content increases the share of the larger pores in the cement matrix. These changes in the pores’ structure also explain the fall in concrete’s compressive strength at the high FBCFA content. 

## 4. Conclusions

Two series of concrete mixes with large and medium binder content (530 and 400 kg/m^3^) were selected for testing. Besides the strength tests, which are the basis for UWC designing, the investigation focused on the effect of FBCFA addition on the rheological properties of UWC mixes. Considering the underwater concreting technology, meeting the requirements regarding the consistency and wash-out loss is obligatory and decides on the future mechanical performance of the hardened underwater concrete. 

Due to the complexity of UWC mixes’ composition and the use of two acting opposite admixtures, superplasticizer and viscosity modifier, the problem of using FBCFA as the cement’s substitute is difficult. The FBCFA content higher than 30% of the cement mass makes it impossible to design the concrete mix with the consistency and wash-out loss suitable for adequately placing it in the formwork under water. Obtaining the constant-in-time rheological properties of UWC mixes containing a large volume of FBCFA requires using a tremendous amount of superplasticizer, high above the content recommended by the producers, which also increases the cost of the concrete mix. The large volume of FBCFA also worsens the UWC compressive strength and tightness.

Limitation of the binder content in UWC mix to 400 kg/m^3^, with the use of up to 30% FBCFA in the composition of the binder, enables the production of UWC with the compressive strength close to the reference concrete and a consistency suitable for placing the concrete mix under water. The highest compressive strength after 56 days was achieved by concretes with 20% FBCFA content. The strength of these concretes was on average 10% higher than that of concrete without the addition of FBCFA. The advantages of FBCFA addition are an increase in the tightness of concrete and the improvement of corrosion resistance.

## Figures and Tables

**Figure 1 materials-15-04809-f001:**
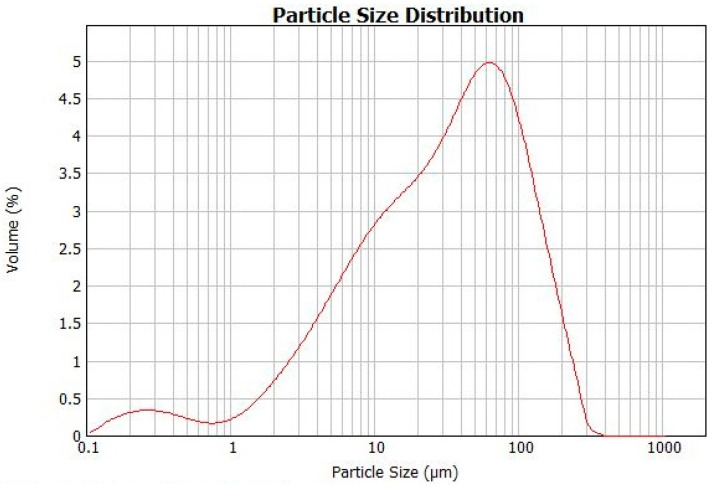
Particle size distribution related to the volume fraction of FBC FA determined using Malvern MasterSizer 2000.

**Figure 2 materials-15-04809-f002:**
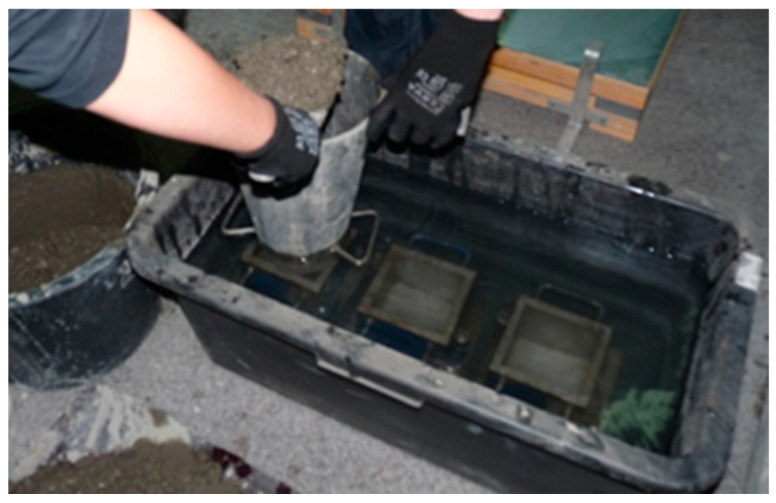
Preparation of specimens for strength testing.

**Figure 3 materials-15-04809-f003:**
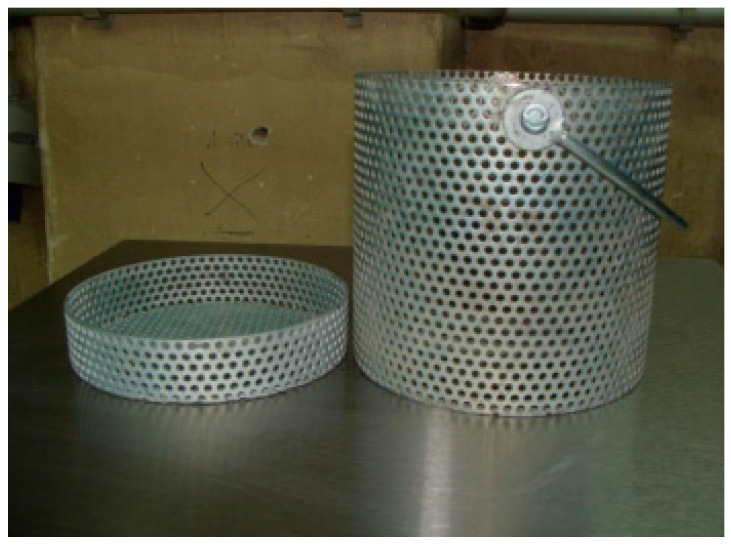
Concrete mix container for wash-out loss testing.

**Figure 4 materials-15-04809-f004:**
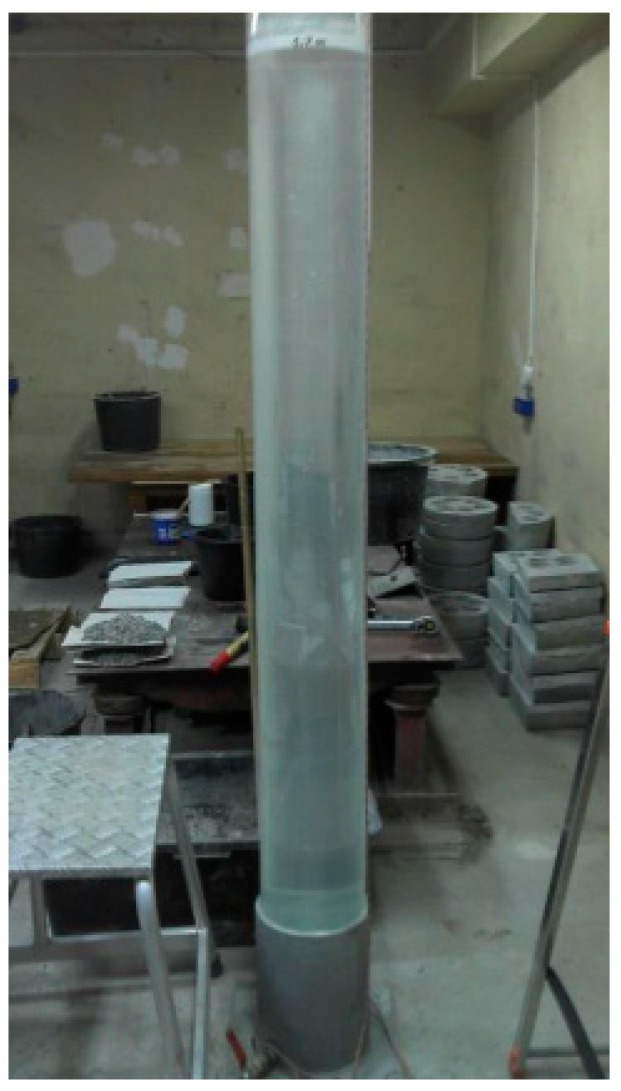
The pipe for wash-out loss testing.

**Figure 5 materials-15-04809-f005:**
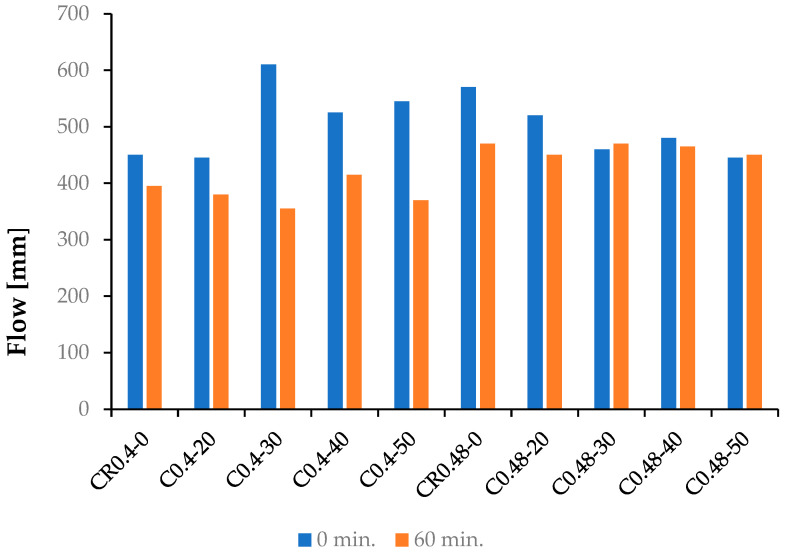
Results of the UWC mixes consistency testing after 0 and 60 min from mixing.

**Figure 6 materials-15-04809-f006:**
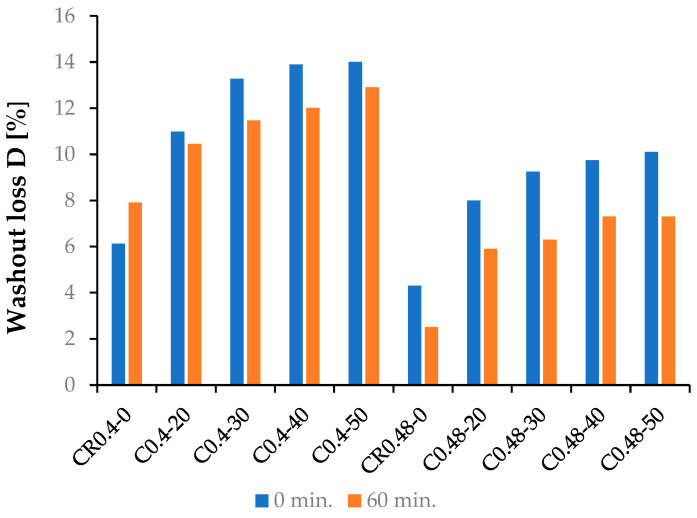
Results of wash-out loss test.

**Figure 7 materials-15-04809-f007:**
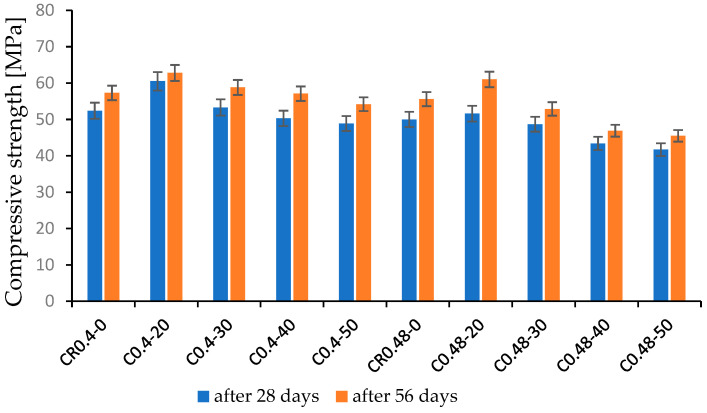
Test results of compressive strength of UWC specimens.

**Table 1 materials-15-04809-t001:** Grain size distribution curves of the sand and gravel.

Aggregate	Sieve [mm]/Remains on the Sieve [%]								
	0.125	0.25	0.5	1	2	4	8	16	31.5
Sand 0/2 mm	13.6	0.3	47.7	26.3	4.8	6.7	0.6	0	0
Gravel 2/8 mm	0	0.9	0.9	7.9	3.5	35.1	49.4	2.3	0
Gravel 8/16 mm	0	0.1	0.2	0.2	0.2	0.4	9.3	84.8	4.8

**Table 2 materials-15-04809-t002:** Chemical composition of fly ashes and cement.

Compound	Content, Mass %	
	FBCFA	Cement
SiO_2_	39.06	21.07
Al_2_O_3_	21.01	5.11
CaO	10.74	63.77
SO_3_	6.83	3.2
Fe_2_O_3_	5.55	4.12
K_2_O	1.98	0.37
MgO	1.87	0.58
Na_2_O	0.54	0.05
Cl	0.12	0.07
TiO_2_	0.80	-
P_2_O_4_	0.64	-
Mn_3_O_4_	0.04	-
TiO_2_	0.80	-
Loss of ignition	10.47	1.66

**Table 3 materials-15-04809-t003:** Concrete mixes compositions.

Concrete Designation	Water/Binder w/b	Cement	FBC FA	Water	Sand 0/2 mm	Gravel 2/8 mm	Gravel 8/16 mm	AWA	SP
		Content [kg/m^3^]							
CR0.4-0	0.40	530	0	212	593	600	428	5.3	5.4
C0.4-20		424	106	212	593	600	428	5.3	8.2
C0.4-30		371	159	212	593	600	428	5.3	11
C0.4-40		318	212	212	593	600	428	5.3	14.5
C0.4-50		265	265	212	593	600	428	5.3	17
CR0.48-0	0.48	400	0	192	593	342	769	4.0	8
C0.48-20		320	80	192	593	342	769	4.0	10
C0.48-30		280	120	192	593	342	769	4.0	12
C0.48-40		240	160	182	593	342	769	4.0	14
C0.48-50		200	200	182	593	342	769	4.0	16

**Table 4 materials-15-04809-t004:** Results of UWC mixes’ consistency and wash-out loss testing.

Mix Designation	Flow [mm]		Washout Loss [%]	
	0 min	60 min	0 min	60 min
CR0.4-0	450	395	6.12	7.91
C0.4-20	445	380	10.99	10.45
C0.4-30	610	355	13.27	11.47
C0.4-40	525	415	13.89	12.01
C0.4-50	545	370	14.01	12.90
CR0.48-0	570	470	4.30	2.50
C0.48-20	520	450	8.00	5.90
C0.48-30	460	470	9.25	6.30
C0.48-40	480	465	9.75	7.30
C0.48-50	445	450	10.10	7.30

**Table 5 materials-15-04809-t005:** The results of UWC compressive strength testing.

Concrete Designation	Compressive Strength [MPa]	
	After 28 Days	After 56 Days
CR0.4-0	52.4 ± 2.0	57.3 ± 2.4
C0.4-20	60.5 ± 2.3	62.8 ± 2.7
C0.4-30	53.3 ± 2.1	57.8 ± 2.4
C0.4-40	50.3 ± 2.3	57.1 ± 2.2
C0.4-50	48.9 ± 1.7	54.2 ± 2.4
CR0.48-0	50.0 ± 1.9	55.6 ± 2.6
C0.48-20	51.6 ± 2.3	61.0 ± 2.8
C0.48-30	48.7 ± 1.8	52.9 ± 2.3
C0.48-40	43.4 ± 1.6	46.9 ± 1.8
C0.48-50	41.7 ± 1.5	45.5 ± 1.6

**Table 6 materials-15-04809-t006:** Water penetration depth of UWC.

Concrete Designation	Water Penetration Depth [mm]			
	No. of Specimen			Max. Value
	1	2	3	
CR0.4-0	19	22	18	22
C0.4-20	12	16	12	16
C0.4-30	13	14	13	14
C0.4-40	12	16	11	16
C0.4-50	18	14	10	18
CR0.48-0	22	20	26	26
C0.48-20	21	13	18	21
C0.48-30	12	11	15	15
C0.48-40	14	16	12	16
C0.48-50	14	18	18	18

## Data Availability

The data presented in this study are available on request from the corresponding author.

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
