# Peer review of "Use of Fluidized Bed Combustion Fly Ash as a Partial Substitute for Cement in Underwater Concrete Mixes"

_materials, 2022, doi:10.3390/ma15144809_

Round 1

Reviewer 1 Report

Journal of Materials

Technical, grammatical, and common mistakes are as follows

Comments for authors;

Ø Write keywords in alphabetical order.

Ø In the introduction part; Up to the temperature of 300÷350°C, 500÷550°C, revise it and make it clear.

Ø Write abbreviation before UWC.

Ø Section 2.2. After 15 s, the container was pulled out within a time not longer than 5±1 s. Revise it.

Ø Section 3.1. mention the UWC mixes instrument and technique.

Ø %, °C, Figure, Table, Heading, Sub-heading, Numbers etc., write the same format throughout the manuscript.

Ø Section 3.2. testing results after 28 and 56 days of curing, mentioned the condition.

Ø Section 3.3. Revise it with more explanation.

Ø Conclusion. Make it short and meaningful.

v Cite the following references

Ø https://doi.org/10.1002/app.51191

Ø 1-10. https://doi.org/10.1002/app.50515

Author Response

Dear Reviewer,

thank you for your valuable comments concerning our manuscript entitled “Use of fluidized bed combustion fly ash as a partial substitute for cement in underwater concrete mixes”  The comments were helpful for improving our paper. We have made corrections ( major changes to the text are marked in yellow), which we hope will find your approval.

Yours sincerely,

Elżbieta Horszczaruk, on behalf of the authors

Reviewer 2 Report

Reducing waste materials is our main concern all over the world. Detailed experimental results (flow, washout loss, and compressive strength) of UWC are revealed and they looks reasonable. This paper utilized FA (below certain amount) as the subitute of cement and can increase the compressive strength of the UWC. However, it still have minor problems need to be clarified. So minor modifications is required.

Comments are as follows,

1 The standard deivations of compressive strength are shown in Table 5. However, they also be shown in Fig. 7.   

2.The decreased compressive strength by high content of the FBC FA might be caused by poor dispersion or aggregation (beyond saturation) of FBC FA. You might include this on your dissusion.

Author Response

(The authors gave the same response as above.)

Reviewer 3 Report

-Please, decrease plagiarism. It is currently 20%. Decrease to 18-15%. SEE THE ATTACHED FILE

-Place error in the bar graphs to observe the measurement error

-In the abstract and conclusions, include numerical values to highlight the most promising results

Author Response

(The authors gave the same response as above.)
